# Activity Restriction and Hospitalization in Pregnancy: Can Bed-Rest Exercise Prevent Deconditioning? A Narrative Review

**DOI:** 10.3390/ijerph20021454

**Published:** 2023-01-13

**Authors:** Montse Palacio, Michelle F. Mottola

**Affiliations:** 1Senior Consultant, Maternal-Fetal Medicine, Hospital Clínic Barcelona (BCNatal Fetal Medicine Research Center), Universitat de Barcelona, 08028 Barcelona, Spain; 2Institut d’Investigacions Biomèdiques August Pi I Sunyer, 08036 Barcelona, Spain; 3Centre for Biomedical Research on Rare Diseases (CIBERER), 08001 Barcelona, Spain; 4R. Samuel McLaughlin Foundation-Exercise and Pregnancy Laboratory, School of Kinesiology, Faculty of Health Sciences, Department of Anatomy and Cell Biology, Schulich School of Medicine and Dentistry, Children’s Health Research Institute, The University of Western Ontario, London, ON N6A 3K7, Canada

**Keywords:** activity restriction, hospitalization, bed-rest exercise, pregnancy, high-risk, intervention

## Abstract

Evidence suggests that exercise during pregnancy is beneficial to both parent and fetus. However, there are high-risk pregnancy conditions that may warrant hospitalization. In our narrative review, we first describe the clinical implications for activity restriction in pregnancy, the effects of hospitalization, and the impact of bed rest on non-pregnant individuals. We provide examples of a 30 min bed-rest exercise program for hospitalized pregnant patients using the principal of suggested frequency, intensity, time (duration) of activity, and type of activity (FITT) using a resistance tool while in bed. If the individual is able to ambulate, we recommend short walks around the ward. Every minute counts and activity should be incorporated into a program at least 3 times per week, or every day if possible. As in all exercise programs, motivation and accountability are essential. Flexibility in timing of the exercise intervention is important due to the scheduling of medical assessments that may occur throughout the day for these hospitalized patients. Evidence suggests that by improving physical and emotional health through a bed-rest exercise program during a hospitalized pregnancy may help the individual resume demanding daily activity in the postpartum period and improve quality of life once birth has occurred. More research is necessary to improve the health of those individuals who are hospitalized during pregnancy, with follow up and support into the postpartum period.

## 1. Introduction

International guidelines [1] state that physical activity is beneficial during pregnancy for both parent and fetus. The results of systematic reviews identified that, compared with no physical activity, accumulating at least 150 min of moderate-to-vigorous intensity physical activity (MVPA) over three or more days per week was associated with clinically meaningful reductions in the odds of developing gestational diabetes mellitus, pre-eclampsia, and gestational hypertension [2]. Exercise should incorporate a variety of aerobic exercise and resistance training activities to achieve greater benefits and some programs detail a supervised exercise routine [3] Adding yoga and/or gentle stretching may also be beneficial. Pelvic floor muscle training exercises (e.g., Kegel exercises) may be performed daily to reduce the odds of urinary incontinence [4].

In these recent guidelines, it has also been stated that individuals with absolute contraindications because of complications of pregnancy (i.e., preeclampsia or preterm labor), may continue their usual activities of daily living but should not participate in more strenuous activities. However, they acknowledge the lack of studies evaluating the effects of different types of physical activity/exercise intervention on these and other medical conditions, underlying the importance of some type of activity over none at all and urge for future research [5].

There is a growing body of evidence that suggests that a sedentary lifestyle during pregnancy puts the individual at risk for chronic disease. An extreme form of sedentarism is bed rest, and as the importance of being physically active during pregnancy grows, the ineffectiveness of restricting activity on preserving pregnancy outcomes is called into question [6,7,8]. If hospitalization of pregnant individuals due to a high-risk pregnancy is prescribed then are there safe ways to increase energy expenditure while in hospital?

In our narrative review, we first describe the clinical implications for activity restriction in pregnancy, the impact of bed rest on non-pregnant individuals, and the effects of hospitalization during pregnancy, followed by the impact of exercise to prevent side effects of activity restriction and a bed-rest exercise program that may reduce the physiological deconditioning in pregnant hospitalized individuals. Urgent research is needed to confirm the benefits of a bed-rest exercise program in activity-restricted patients who have high-risk pregnancies.

## 2. Activity Restriction in Pregnancy

Despite the fact that restriction of physical activity has been discouraged by well-known international institutions [9], individuals who develop complications during pregnancy have traditionally been prescribed activity restriction (the most severe form being complete bed red) which could lead to a sedentary lifestyle. Indeed, even though highlights from comprehensive meta-analyses have suggested no benefits of bed rest on pregnancy outcomes [10,11,12], up to 87% of North American clinicians still prescribe bed rest for high-risk pregnant individuals for lack of better therapeutic options [12]. Of specific concern, is that the majority of those physicians prescribing bed rest during pregnancy do not offer any recommendations for rehabilitative exercise [13], which would lessen the negative effects of activity restriction [14].

In the clinical field, despite current suggestions against recommending bed rest, activity restriction remains a common therapy used to prevent preterm birth in several medical settings. In addition, hospital admission or a strong recommendation for activity restriction are the common interventions for preterm premature rupture of membranes, vaginal bleeding with or without placenta previa, multiple gestation, hypertensive disorders of pregnancy, short cervical length, and fetal growth restriction, despite the growing body of literature which suggests that activity restriction does not prevent adverse perinatal outcomes, but may exacerbate physical and psychosocial risks [15].

Indeed, there is information suggesting that approximately one third of high-risk pregnant individuals were placed on activity restriction despite a departmental policy against it [16]. The 100% compliance rate in patients placed on activity restriction is a strong reminder of the physician impact that prescribing patterns have on patients [16]. A mail-based survey of all Society for Maternal-Fetal Medicine members in the United States asking whether they would recommend bed rest in the setting of arrested preterm labor or preterm premature rupture of membranes (PPROM) at 26 weeks shows that most maternal–fetal medicine specialists recommend bed rest despite the belief that bed rest is associated with minimal or no benefit. Bed rest was defined as no more than 1–2 h per day out of bed, with permitted activities including bathroom use, bathing, and brief ambulation inside the home/hospital. Randomized, prospective trials are still necessary to evaluate the efficacy of bed rest in these settings [17].

Therefore, “therapeutic” bed rest continues to be used widely in the clinical field, despite evidence of no benefit and known harms. Cochrane systematic reviews do not support “therapeutic” bed rest for threatened abortion, hypertension, preeclampsia, preterm birth, multiple gestations, or impaired fetal growth [12,15]. This Cochran assessment is echoed in other systematic reviews. Bed rest prescription is inconsistent with the ethical principles of autonomy, beneficence, and justice. Thus, if bed rest is to be utilized, it should be only in the context of a formal clinical trial [12].

However, when severe complications occur and the pregnant individual needs admission to hospital, bed rest is not discouraged, and any kind of mobility or physical activity might be severely restricted for days or weeks. Therefore, ‘usual activities of daily living’ are not possible, and activity is dramatically restricted, leaving these individuals to total inactivity that may preclude the benefit of adequate physical exercise.

## 3. Methods

Surprisingly, although studies evaluating the impact of bed rest or immobility in other contexts (e.g., medical conditions requiring intensive care unit admission, spinal cord injuries, experiments in degravitation in astronauts) have been conducted, there exists scarce research in the literature on the evaluation of adapted physical activity in those individuals to whom bed rest is prescribed. For our narrative review, we found 526 entries in PubMed for ‘physical activity’ AND ‘pregnancy’, only 80 for ‘hospitalization’ AND ‘pregnancy’, 50 for ‘bed rest’ AND ‘pregnancy’, and 3 for ‘activity restriction’ AND ‘pregnancy’. We collated the available literature and summarized our important findings.

## 4. Effects of Hospitalization and Bed Rest in Pregnancy

Effects of bed rest and micro-gravity have been studied in non-pregnant individuals by the National Aeronautics and Space Administration (NASA) and have extensively described a series of side-effects for every system, which included fatigue, headache, mood changes, lower back pain, tenseness, difficulty concentrating, back muscle soreness, dry skin, cardiac atrophy, augmented heart rate, blood coagulation, heartburn and reflux, constipation, diminished lung compliance, decreased cardiac output and stroke volume, muscle atrophy, joint contracture, thromboembolic disease, skin ulcers, glucose intolerance, skeletal muscle insulin resistance, and increased concentrations of blood cholesterol and triglycerides [18]. This suggests that bed rest might be a more potentially harmful treatment than a benefit for most medical conditions [19]. Indeed, it is a long list and many of these side-effects have also been detected in activity-restricted pregnant individuals who have been hospitalized due to a high-risk pregnancy.

Scarce data exist on the amount of physical activity performed by pregnant individuals while in hospital. Data available suggest that hospitalized pregnant individuals self-impose bed rest even in the absence of a medical recommendation [20]. Studies on ambulatory third trimester pregnant individuals show a step count of 6495, hospitalized with partial or no bed rest prescribed, show that these individuals perform about 1300–1500 steps and indeed, the level of step counts in those with strict bed rest drops dramatically to 378 steps [21,22,23,24]. Since the activity levels of hospital-admitted pregnant individuals are drastically reduced, many show similar physiological side-effects to non-pregnant bed-rested individuals. These symptoms can include cardiovascular deconditioning (loss of physical strength through illness, injury, or no activity), muscle atrophy, bone demineralization, and decreased weight gain [22,24,25,26].

As physical (hormonal, respiratory, cardiovascular, and skeletal) modifications and psychosocial (stress, anxiety, and depression) changes occur in normal pregnancy, pregnant individuals who are restricted in their daily activity or confined to bed rest due to complications, may potentially present even more deep alterations that may jeopardize their ability to resume a demanding activity in the postpartum period. In fact, activity restriction or bed rest does not appear to mediate pregnancy complications and is associated with negative maternal physiological and psychosocial outcomes, while not improving fetal health [8,25].

In non-pregnant individuals, bed rest is related to an increase in heart rate and vascular resistance and a decreased stroke volume [27], which suggest cardiovascular deconditioning and a potential risk for future cardiovascular disease. These cardiovascular changes may also occur in the hospitalized pregnant individual. As an additional consequence of deconditioning, pregnant individuals who have been severely activity-restricted may have more difficulties resuming simple mobility tasks during the first week postpartum (such as walking upstairs), as their lower limbs might be unable to bear their weight. This is due, in part, to the lost muscle mass that occurs from lack of use while in hospital which may jeopardize prompt ambulation [25,26]. This also increases the risk for first venous thromboembolism, which is increased during admission to hospital and is not related to delivery, but remains significantly higher in the 28 days after discharge [28].

Muscle loss has been reported after 5 days of bed rest. Standard measurements for the cross-sectional area of both the calf and thigh muscles have shown a 2–3 percent decline [29]. Reduction in skeletal muscle mass due to activity restriction may be problematic during pregnancy, because skeletal muscle contraction is important in maternal blood glucose regulation [30]. In non-pregnant individuals, as little as three days of bed rest may induce insulin resistance, glucose intolerance, and hyperglycemia [31]. Thus, a combination of activity restriction for an extended period of time with little muscular activity and loss in muscle mass during pregnancy may place these individuals at risk for developing gestational diabetes mellitus [21,32].

Hospitalized activity restriction is also related to maternal weight loss [33]. During the first week of bed rest, 75.2% of individuals (*n* = 106) either lost or failed to gain weight. As fetal weight is related to maternal weight gain, it is uncertain if the higher small-for-gestational age rate described in pregnant individuals who are prescribed hospital bed rest is due to the level of activity restriction, or simply because of the high-risk nature of the pregnancies to whom bed rest was prescribed. Although maternal weight loss is likely a synthesis of several issues, including muscle atrophy, this suggests that efforts to maintain maternal weight gain during bed rest is worthwhile.

Bone health is important to evaluate in patients who are restricted from weight-bearing activities for long durations. Pregnant individuals placed on bed rest are six times more likely to lose more than 5% of their bone density within 20 weeks [34]. Hospitalized pregnant individuals (average hospital stay of 16 days), with only an average of 1504 ± 1377 steps per day, presented significant differences in bone stiffness index scores compared to pregnant individuals who were ambulatory [24]. An additional study showed that urinary deoxypyridinoline excretion, a marker of bone resorption, was higher in hospital-admitted pregnant individuals compared to those of ambulatory participants, indicating that an increased bone resorption occurs in hospitalized pregnant individuals. Indeed, deoxypyridinoline excretion was even more increased the longer an individual was activity restricted. This would suggest that a reduction in weight-bearing activities for two to three weeks may an impact on ensuing bone health, increasing the risk of future osteopenia, osteoporosis and the risk of fracture [22].

Bed rest has also been associated with sleep alteration and psychosocial side-effects. Sleep–wake cycles are altered, shifting diurnal rhythms (patterns of sleep during day and night) in a way that even though overall sleep time is similar, daytime sleep is increased while nighttime is decreased. Therefore, prolonged antepartum hospitalization has a negative impact on sleep duration and quality [35]. Regarding the psychosocial area, activity restriction can be accompanied by a myriad of side-effects that may be present both during pregnancy and the postpartum period: depression, anxiety, stress, and boredom might be present [36,37]. While in hospital, the pregnant individual is often separated from family, friends, and the job setting, and may have difficulty dealing with the additional worry of a high-risk pregnancy. Individuals who were prescribed bed rest in hospital or at home had higher stress scores relating to thinking about babies’ health, feeling dependent on others, and feeling uncertain about the outcome [38]. Overall, the severity of these side-effects appears to be directly related to the degree of activity restriction [39].

## 5. Exercise to Prevent Side Effects of Activity Restriction

It is evident from the literature that prolonged bed rest has detrimental side-effects. As muscle atrophy can be objectively determined in as little as four to five days of bed rest, it is imperative to administer measures of prevention such as an in-bed exercise program or short periods of ambulation if the patients are able to do so [40].

There are studies that suggest exercise prescription may counteract deconditioning in non-pregnant females on bed rest. In this population, resistance exercise and passive stretching of muscles may be a successful countermeasure to diminish cardiovascular deconditioning [27,41] and bone deterioration accumulated during bed rest [29,42,43,44]. Literature has suggested that a combined resistance and aerobic exercise program prevents losses in strength, endurance, and protects lower limb muscle mass, whereas initiating a dietary intervention only may not be effective in preventing these side-effects [45].

In healthy pregnant individuals, systematic reviews found that ‘mixed’ interventions which combine aerobic and resistance training activities showed greater improvements in pregnancy outcomes than aerobic activity alone [1]. While there has been fewer research studies on resistance exercises as compared to aerobic exercises in pregnancy, available data regarding resistance exercise in pregnancy has not identified harmful effects on the participant, the fetus, or the neonate [3]. A warm-up and cool-down phase should be included in any physical activity program. Ligaments become relaxed during pregnancy because of increasing hormone concentrations and may influence the range of movement, thereby increasing injury risk. Exercise should incorporate a variety of aerobic exercise and resistance training activities to achieve numerous benefits. Adding yoga and/or gentle stretching may also be advantageous. Pelvic floor muscle training (exercises (e.g., Kegel exercises) may be executed on a daily basis to reduce the odds of urinary incontinence [1].

Regarding high-risk pregnant individuals who are activity-restricted, some studies have described exercises that may be performed for individuals on bed rest. First, the recommendation for bed rest should be re-considered: those at risk for preterm labor may be able to partake in an exercise program with negligible risk of increasing uterine activity [46]. An aquatic exercise program showed some benefits to hospitalized pregnant individuals [47]. If bed rest is finally prescribed, or mandatory because of the need for continuous maternal monitoring, a described bed-rest exercise program is shown to be safe for both the pregnant individual and the fetus, with no associated adverse outcomes or increasing uterine activity [48]. This consisted of a 30 min three-phase exercise program using resistance bands as a form of resistance exercise for both lower and upper body with 5 min stretching for a warm-up and cool-down phase, and 20 min of muscle strengthening exercises. See Figure 1 as an example of a bed-rest exercise program. These exercises can be modified to the needs of the pregnant individual. In addition, the frequency, intensity, duration, and type of exercise [49] used in a hospitalized bed-rest exercise program is important to consider. The program must be flexible to accommodate the constantly changing schedule of these patients due to the timing of medical tests throughout the day. However, as in any exercise program, it is important to keep the individual motivated and accountable for continuing the program while hospitalized and to also assist in maintaining the program once birth has occurred, in order to help with rehabilitation from activity restriction in the postpartum period.

Although it is suggested that the bed-rest exercise intervention be performed at least three times per week [48], it can be performed daily if timetables allow. If 30 consecutive minutes are not possible, then perhaps dividing the exercise program into smaller segments whereby only lower body exercises are conducted for 10 min and upper body exercises may be conducted later that day when time permits will also work. Short warm-up and cool-down exercises should always accompany even short bouts of the program. The intensity of the muscle conditioning bed-rest exercise program depends on the strength of the individual and the resistance tool should be modified to accommodate the ability of the participant. For example, a resistance band comes in different stretch tensions and an individual who is just starting the program should use a resistance band with the least amount of tension and work up to increasing tension, as this will make the exercises more difficult. If the individual is able to take short walks around the ward in addition to the bed-rest exercise program of resistance and muscle strengthening, this is also recommended. Gentle core exercise might be included as well if there is no specific contraindication. Follow-up into the postpartum period is also recommended to assist the hospitalized individual to rehabilitate to pre-pregnancy fitness levels.

### Need for Future Study

Unfortunately, poor attention has been paid to the fact that pregnant individuals who have been activity-restricted have special needs to resume their activity once discharged from hospital. Few studies have evaluated an intervention to reduce the side-effects of activity restriction in hospitalized pregnant individuals, even though resistance exercise interventions have been shown to be effective in non-pregnant populations [29,42]. Individuals are discharged from prolonged hospitalization after giving birth without the resources to aid in recuperating from the physiological deconditioning that occurs with lengthy stays in hospital [26]. Antepartum activity-restricted hospitalized individuals are expected to return home shortly after childbirth to take on the responsibility of tending to a new infant, in addition to other children and household responsibilities. Indeed, care of the newborn overshadows the needs of the gestational parent. In addition, if the baby or babies are high-risk and are admitted to the neonatal intensive care unit, there are no programs in place to assist the postpartum individual or their partner in coping with the after-effects of prolonged pregnancy hospitalization while also dealing with hospitalized high-risk infant(s). This is an extremely stressful time for the individual and their family, and the need for structured activity rehabilitation programs to accommodate previously hospitalized activity-restricted individuals to improve mental health and overall wellbeing are vital.

Therefore, efforts are needed to improve knowledge in this area. A potential solution to the side-effects of activity restriction may be to incorporate exercise or some form of activity into the routines of hospitalized pregnant individuals to lessen the physiological and psychosocial side-effects of activity restriction and to assist them in continuing activity into the postpartum period. The message for both physicians and pregnant individuals under activity restriction is to ‘Modify, don’t stop!’ and to reconsider how to respond to the ‘relative’ and ‘absolute’ contraindications to physical activity in pregnancy [8]. More research is vital to improve the health of those individuals who are hospitalized during pregnancy, with follow up into the postpartum period.

## 6. Conclusions

Despite potential benefits, even at activity levels below current guidelines, it is crucial that the frequency, intensity, duration, and/or volume of exercise be modified for those with contraindications, rather than preventing overall physical activity [14]. Even though standard exercise should aim to achieve 150 min of MVPA spread over at least 3 days weekly, substantial benefits are also detected at volumes below these recommendations. It has been shown that engaging in some physical activity is better than none. For many outcomes, even lower intensity physical activity also imparts benefits and, therefore, ‘every minute counts’. Pregnant individuals should be inspired to be physically active, even if they are unable to meet these guidelines [1].

As physical activity includes a range of movement behaviours, including moderate-to-vigorous physical activity, or light activities (e.g., gentle walking and activities of daily living) [14], we need to find those activities that should be prescribed to pregnant individuals who, for some reason, are bed-rested or have restricted activity, to prevent deconditioning, and reduce sedentary behaviors (e.g., sitting/screen time). A recent study produced a list of the top 10 research areas where priorities and issues regarding physical activity, sedentary behavior, and sleep in pregnancy have been highlighted [50]. Indeed, meeting the minimum physical activity recommendations is associated with better perceived health, and a reduction in the prevalence and severity of depression and anxiety at both mid-pregnancy and later pregnancy [51]. Overall, improving physical and emotional health while pregnant will help resume demanding daily activity in the postpartum period, and improve the quality of life for those who are hospitalized and activity-restricted [52,53].

## Figures and Tables

**Figure 1 ijerph-20-01454-f001:**
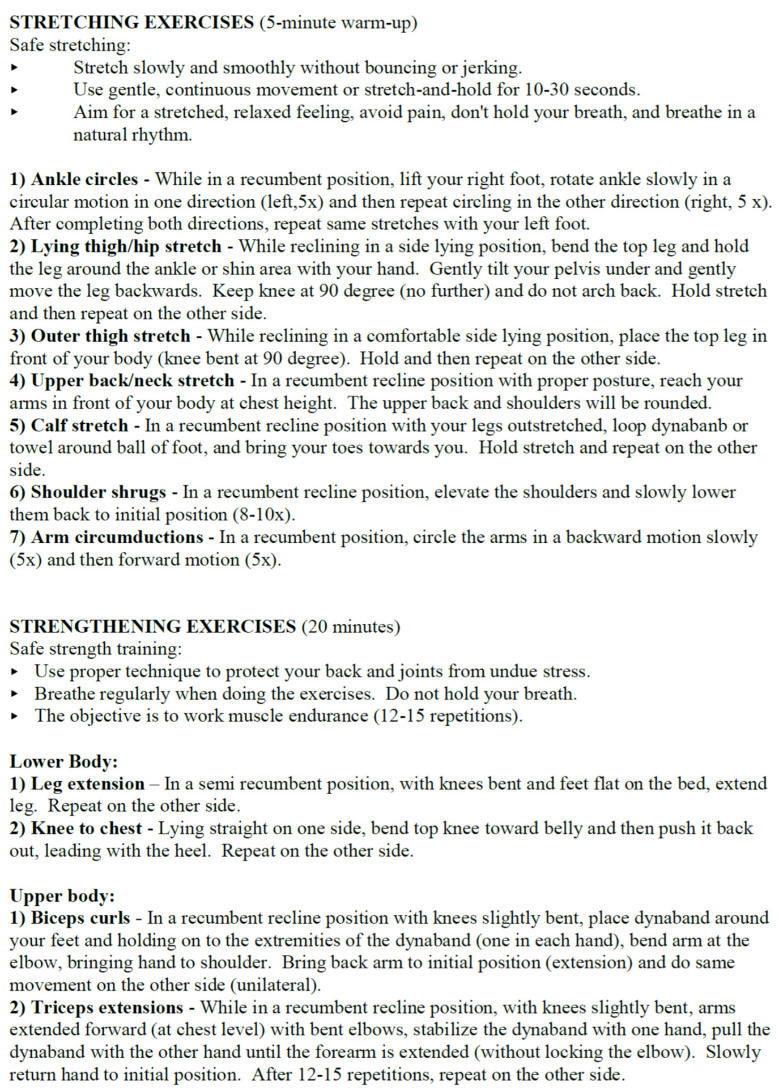
Example of a 30 min bed-rest exercise intervention using a resistance tool (band) for hospitalized pregnant individuals [48].

## Data Availability

Not applicable.

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
