# Peer review of "Activity Restriction and Hospitalization in Pregnancy: Can Bed-Rest Exercise Prevent Deconditioning? A Narrative Review"

_ijerph, 2023, doi:10.3390/ijerph20021454_

Round 1

Reviewer 1 Report

Manuscript # IJERPH-1979389

The manuscript voices a strong but important opinion about the prevention of cardiovascular deconditioning associated with strict bed rest associated with high risk conditions during pregnancy., a noble and needed opinion.

I have several issues which when addressed will improve the opinion piece.

1.  The manuscript does not reflect a the established standards for a systematic review.

2. There is a lack of definition of cardiovascular deconditioning in pregnant women. the cardiovascular physiology in normal pregnancy is dynamic by trimester and different in strict left-sided bedrest.

a. Does deconditioning occur in normal fully active  women through pregnancy?
b. The comparison to microgravity effects in mainly male astronauts is not valid. 
c. While the normal mother and fetus are able to regular excercises up to a maximum of 70% VO2  up to 150 mins a week as predicted by their pregnancy physiology, there is little research on the pregnancy physiology in cardiovascular deconditioning periods. The manuscript does  not review what scientific study that exists.

2. The manuscript fails to  adequately fetal/ neonatal risk in the absence of strict bed rest in certain high risk situations including premature delivery < 32 weeks. ROM, CX DILATION > 4 cm, etc

3. Strict bed rest at a level III MFM/ NICU team in hospital with very high risk conditions is necessary for rapid best care for the mother and fetus,l.e. Severe preeclampsia at < 30 weeks, cord presentation with ruptured membranes < 32 weeks, mildly symptomatic placenta prévia after prior cesaren section, monochorionic  multifetal gestation, vasa previa, etc. A table of these extremely high risk situations would improve the recommendations. 

4. Is there any information on changes in intraabdominal pressure or uterine contraction before, during, and after the recommended exercise routine in figure 1.

5. Caution is warranted when projecting muscle loss after traumatic orthopedic injury to hypothesized muscle mass loss with strict bed rest in the third trimester. Have there been any studies in normal women at bed rest in the third trimester? 
6. The manuscript does not mention the unfortunate risk of legal challenges for a new practice. Given the liability risk, the introduction of this novel, but good idea, should be performed under an IRB research protocol on lower risk populations initially.

In general, the authors are citing the opinions of the authors in other reviews or reports. For better reader clarity, it would be valuable to cite the statistics of the research article that best illustrates the point hat the current authors would like to make.

1. While it is established that regular exercise to 50-70% maxVO2 for at least 150 minutes in healthy pregnant women and feti, and has the potential to reduce diabetic and hypertensive disease, the manuscript has limited review of the physiology of deconditioning in pregnancy in a scientific studies. Using data from microgravity studies in primarily male well conditioned astronauts  is inappropriate support.

2. There is no review of the fetal/neonatal effects of strict bed rest. The two dominant issues for the child is preter delivery

Reviewer 2 Report

It is a well-argued article that has a clear motivation for being written. In reality, this article deserves to be published, but:

1. It has no methodology. It sounds more like a book chapter than an article to be published. What type of article is it and what Methodology and Materials does it have?

2. Some parts and chapters are too long (including the conclusions) and too wordy, they should be shortened and the language more precise.

3. I think it deserves to be published, but the academic rigor definitely deserves to be improved.

Reviewer 3 Report

This is an elegant review of the harmful effects of severe physical activity restriction in high-risk pregnancies and position stand on the need to develop strategies to remedy the harmful consequences of inactivity in this population. Offering a pleasant and rather comprehensive description of the physiological, psychological and behavioural negative effects of bed rest and severe activity restriction on pregnant women, the review also describes in practical sense guidelines for rehabilitative intervention and for prescription of physical activity based on general principles that can be adapted to high-risk pregnancy. Surely, this is an important topic that is of interest to a large audience. Therefore, I’m pleased to endorse the publication of the manuscript.

Round 2

Reviewer 1 Report

This is an opinion article not a balanced scholarly review review.

Author Response

We agree that our manuscript does contain opinion but we also have reviewed the scant literature on this topic which is why we have added "narrative review" to the title as well as throughout the manuscript. 

Reviewer 2 Report

Thank you for your answer.

However, I still do not believe that the manuscript is in the final form for being published.

It still needs a separate section with Methods, even if it is a narrative review. It is not enough to mention that it is a narrative review. 

I kindly asked for a short form of the manuscript, however, it is still too large.

Author Response

Reviewer#2 - We thank you for your comments. Our responses are in bold.

1) It still needs a separate section with Methods, even if it is a narrative review. It is not enough to mention that it is a narrative review.

We have added the title of METHODS in the manuscript as suggested.

2) I kindly asked for a short form of the manuscript, however, it is still too large.

We checked the number of words required for the manuscript from the “Instructions for authors” and found that for submitted papers there is a requirement of 3000 words minimum and for a review article there is a requirement of  4000 words minimum. Our manuscript as it stands has 3477 words in the text body and meets the requirements of the journal. We do not believe that we should cut any more words from our manuscript as this will diminish the content and the meaning of our paper.